# Facile Synthesis and Characterization of Cupric Oxide Loaded 2D Structure Graphitic Carbon Nitride (*g*-C_3_N_4_) Nanocomposite: In Vitro Anti-Bacterial and Fungal Interaction Studies

**DOI:** 10.3390/jof9030310

**Published:** 2023-02-28

**Authors:** Rajendran Lakshmi Priya, Bheeranna Kariyanna, Sengodan Karthi, Raja Sudhakaran, Sundaram Ganesh Babu, Radhakrishnan Vidya

**Affiliations:** 1Department of Chemistry, School of Advanced Sciences, Vellore Institute of Technology, Vellore 632014, Tamil Nadu, India; 2VIT School of Agricultural Innovations and Advanced Learning, Vellore Institute of Technology, Vellore 632014, Tamil Nadu, India; 3Department of Entomology, College of Agriculture, Food and Environment, University of Kentucky, Lexington, KY 40503, USA; 4School of Biosciences and Technology, Vellore Institute of Technology, Vellore 632014, Tamilnadu, India

**Keywords:** antibacterial, antifungal activity, CuO/g-C_3_N_4_ nanocomposite, fungal interactions, plant pathogens

## Abstract

The active and inexpensive catalyst cupric oxide (CuO) loaded foliar fertilizer of graphitic carbon nitride (*g*-C_3_N_4_) is investigated for biological applications due to its low cost and easy synthesis. The synthesized CuO NPs, bulk g-C_3_N_4_, exfoliated *g*-C_3_N_4_, and different weight percentages of 30 wt%, 40 wt%, 50 wt%, 60 wt%, and 70 wt% CuO-loaded *g*-C_3_N_4_ are characterized using different analytical techniques, including powder X-ray diffraction, scanning electron microscopy, energy dispersive X-ray analysis, and ultraviolet-visible spectroscopy. The nanocomposite of CuO NPs loaded *g*-C_3_N_4_ exhibits antibacterial activity against Gram-positive bacteria (*Staphylococcus aureus* and *Streptococcus pyogenes*) and Gram-negative bacteria (*Escherichia coli* and *Pseudomonas aeruginosa*). The 20 μg/mL of 70 wt% CuO/*g*-C_3_N_4_ nanocomposite showed an efficiency of 98% for Gram-positive bacteria, 80% for *E. Coli*, and 85% for *P. aeruginosa*. In the same way, since the 70 wt% CuO/*g*-C_3_N_4_ nanocomposite showed the best results for antibacterial activity, the same compound was evaluated for anti-fungal activity. For this purpose, the fungi *Fusarium oxysporum* and *Trichoderma viride* were used. The anti-fungal activity experiments were not conducted in the presence of sunlight, and no appreciable fungal inhibition was observed. As per the literature, the presence of the catalyst *g*-C_3_N_4_, without an external light source, reduces the fungal inhibition performance. Hence, in the future, some modifications in the experimental conditions should be considered to improve the anti-fungal activity.

## 1. Introduction

Different types of microorganisms cause disease in human beings. Recent years have seen an increase in the number of alternative antimicrobial agents or drugs (such as metal nanoparticles, polymers, and peptides) for the treatment of antibiotic-resistant infections [1]. In this study, we focused on the synthesis, characterization, antimicrobial activity, and interactions of fungi. Microorganisms produce infections in humans, animals, and plants. These infections can cause unexpected death in humans. Major causes of death, such as those related to the respiratory system, the gastrointestinal system, the central nervous system, etc., are frequently caused by bacterial infections. One of the bacteria of *P. aeruginosa* can cause serious lung problems, and it can be difficult to treat the disease due to resulting bloodstream infections. A few studies reported a rate of *P. aeruginosa* infection of up to 61%. While most studies consider plant diseases to be of great importance, bacteria-induced diseases in humans, such as cholera, diphtheria, dysentery, plague, pneumonia, tuberculosis (TB), and typhoid, among many others [2], can be deadly.

On the other hand, plant diseases have intensive effects on human civilizations because plant diseases affect food, ornamentals, and natural environments. These plant diseases are caused by pathogens or microorganisms. These pathogens cause plant diseases to spread through fungi, bacteria, and mycoplasmas, and they can easily disperse disease from an infected plant to a healthy plant. Pathogenic fungi cause plant infections such as anthracnose, leaf spot, rust, wilt, blight, coils, scab, gall, canker, damping-off, root rot, mildew, and dieback [3]. Some of the fungi (such as Clubroot, *Pythium* species, *Fusarium* species, *Rhizoctonia* species, and *Sclerotium* species) are responsible for foliar diseases and soilborne diseases [2]. Fusarium pathogens mostly enter through root wounds caused by cultivation. These diseases are effectively controlled by some medicines, but these fungicides are costly and not eco-friendly, motivating the search to find non-conventional alternatives to control plant pathogens.

Different types of metal oxide and metallic nanoparticles were fabricated and used for their antimicrobial activity. Metal oxide in the nano range contains crucial properties that vary by size, chemical composition, and surface chemistry. Transition metal oxide nanoparticles, such as CuO NPs, have been broadly used for various applications, including as sensors, catalysts, semiconductors, antimicrobial agents, supercapacitors, etc. [4,5,6,7]. Additionally, CuO nanoparticles have recently been used in biomedical and cancer applications owing to their attractive chemical properties. Currently, one of the most attractive and active metals is copper, which has a long history in biological chemistry [8,9]. CuO nanoparticles showed excellent antimicrobial activity against different bacterial strains. Previous studies described that CuO nanoparticles exhibited high potential activity for Gram-positive and Gram-negative bacteria, with high efficiency surpassing the accepted therapeutic indices. Researchers have used nanotechnology in agriculture for applications involving agrochemicals, nutrients, pesticides, etc. The uses of nanomaterials are easily adaptable to soil climates and crop conditions. Copper is an essential plant nutrient. It aids in plant growth and disease resistance. In addition, copper is necessary for the creation of key plant defense proteins, peroxidase, and copper multiple oxidases in response to pathogen diseases. The CuO NPs can affect plant nutrition, as demonstrated for the first time by Elmer et al. The Cu_2_O NPs can be used as a nano-fertilizer in plant diseases, effectively suppressing the disease.

Graphitic carbon nitride (*g*-C_3_N_4_) is a stacked morphology of two-dimensional (2D) layered polymeric material composed of sp_2_-hybridized nitrogen-substituted graphene of tris-triazine-based patterns with a small amount of hydrogen. *g*-C_3_N_4_ has a carbon-nitrogen bond, a covalent bond between carbon and nitrogen, and is one of the most plentiful bonds in organic chemistry and biochemistry [10,11]. These materials were prepared by the calcination method. Prepared using the exfoliated *g*-C_3_N_4_ process, the bulk carbon nitride forms by self-condensation of the exfoliation process [12,13]. It exhibits remarkable features including bio-friendliness, tunable electronic structures, good thermal and chemical stability, low cost, simple preparation, and large surface area [14,15]. It can be used for biocompatible medical coatings, chemically inert coatings, drug delivery, catalysis, degradation, sensing, insulators, and supercapacitors [16]. As an antibacterial agent, g-C_3_N_4_ nanosheets possess the ability to produce reactive oxygen species (ROS), followed by metal nanoparticles reacting to the reactive oxygen species (·O_2_^-^ and ·OH), and under low-intensity light irradiation, bacteria were killed [17,18].

In the present study, the synthesis of CuO-loaded g-C_3_N_4_ nanocomposites is achieved using dry synthesis methods. Moreover, the antibacterial activity of the compound is investigated, and the fungal interactions with the compound are also discussed.

## 2. Materials and Methods

### 2.1. Materials

Copper acetate Cu(CH_3_COO)_2_, H_2_SO_4_, and melamine were acquired from Sisco Research Laboratories Pvt. Ltd., Mumbai, India. All the chemicals used in the experimental process were of pure analytical grade and used as such, without any further processing being performed.

### 2.2. Synthesis of g-C_3_N_4_

Initially, the bulk-graphitic carbon nitride (Bulk g-C_3_N_4_) was prepared under atmospheric air conditions by the thermal polymerization of melamine. The temperature was set to 550 °C in a muffle furnace and 6 g of melamine was placed in an alumina crucible for 4 h. After that, the sample was cooled at room temperature. Next, the obtained solid material was ground using a mortar, forming a yellow-colored powder, and the final yield was 3.94 g. During the thermal treatment, due to the decomposition of ammonia, the color changed from intense white to pale yellow. The resultant sample was named bulk-C_3_N_4_. Secondly, the exfoliated graphitic carbon nitride sheets (g-C_3_N_4_) were prepared via the thermal exfoliation method using prepared bulk g-C_3_N_4_. In detail, to the 2 g of prepared bulk g-C_3_N_4_, 20 mL of H_2_SO_4_ was added and stirred for 12 h. Then 200 mL of deionized water was added to the mixture. After that, the resultant mixture was washed with deionized water until it reached a neutral pH. The faint yellow-colored sample was dried at 60 °C for 12 h in a hot air oven. Finally, the faint yellow-colored powder was poured into an alumina boat, and this boat was placed into a tubular furnace at 550 °C (2 °C/min heating rate, for 4 h, under a nitrogen gas flow), and the porous sheet of exfoliated g-C_3_N_4_ was obtained [14] (Figure 1).

### 2.3. Synthesis of Cupric Oxide Loaded g-C_3_N_4_ Nanocomposites

Synthesis of cupric oxide (CuO) loaded graphitic carbon nitride (g-C_3_N_4_) nanocomposites by the direct heating process of the dry synthesis method. In the detailed protocol, various amounts of *g*-C_3_N_4_ and copper acetate compounds were grained for 30 min. No solvent or capping agents were used in this method. The grained mixture was navy-blue in color. Then, the solid mixture was taken in a silica crucible and placed in a furnace for 3 h at 550 °C. The mixture was then cooled to room temperature, and we obtained a brown-colored mixture of CuO/g-C_3_N_4_ nanocomposites as shown in Figure 2: CuO, 30 wt% CuO/*g*-C_3_N_4_, 40 wt% CuO/*g*-C_3_N_4_, 50 wt% CuO/*g*-C_3_N_4_, 60 wt% CuO/ *g*-C_3_N_4_, and 70 wt% CuO/*g*-C_3_N_4_, respectively. The various compounds were compared to the results of the antimicrobial analysis.

### 2.4. Characterization Techniques

The phase and crystallographic structure of as-prepared nanocrystals were observed by a Powder X-ray diffractometer (Bruker, D8 Advance). The diffractograms were recorded for 2Ɵ in the range of 5–80°, with a time of 6 s. The SEM images were captured by scanning electron microscope SEM-ZEISS (EVO18)), Carl Zeiss Microscopy GmbH, 07745 Jena, Germany, before applying spectral coating using gold platinum metal (Quorum). The elemental analysis of the catalyst was analyzed by energy dispersive X-ray (EDX; VEGA3 XUM/TESCAN). UV-Visible absorption spectra were analyzed on JASCO (V-670 PC) equipment, with a wavelength range of 200–1000 nm at room temperature. The optical band gap of these materials was estimated using the Tauc equation.

### 2.5. Anti-Bacterial Activity

#### 2.5.1. Preparation of Inoculum

The antimicrobial properties of the given samples were tested against Gram-positive bacteria (*Staphylococcus aureus* ATCC 25923, *Streptococcus pyogenes* ATCC 19615) and Gram-negative bacteria (*Escherichia coli* ATCC 25922, *Pseudomonas aeruginosa* ATCC 27853). All bacteria were pre-cultured in Mueller Hinton Broth (MHB) in a rotary shaker at 37 °C for 18 h. Next, each strain was modified at a concentration of 10^8^ cells/mL using the 0.5 McFarland standard [19].

#### 2.5.2. Agar Well Diffusion Method

The fresh bacterial culture was pipetted in a sterile petri dish. Molten-cooled Mueller Hinton Agar (MAH) was then poured into the petri dish and blended well. Upon solidification, wells were made using a sterile cork borer (6 mm in diameter) into the agar plates containing the inoculums. Then, 50 μL of the sample (20 μg/μL concentration) was added to the respective wells. These plates were then incubated at 37 °C for 18 h. After the incubation period, the antibacterial activity was obtained by measuring the zone of inhibition (including the diameter of the wall). Saline was applied as a negative control.

### 2.6. Fungal Activity

The antifungal activity of the nanocomposites prepared from a facile dry synthesis method was tested against 2 fungi, namely *Fusarium oxysporum* and *Trichoderma viride.*

#### Agar Well Diffusion Method

The fungal cultures *Fusarium oxysporum* and *Trichoderma viride* were cultured in potato dextrose broth. PDA agar plates were prepared. Four wells were bored in each plate. Overnight fungal cultures were swabbed into the PDA plates. The prepared compounds were pipetted into the wells in volumes of 20, 40, 60, and 80 μL. The plates were incubated at room temperature for 2–3 days to check the formation of the zone around the well. The plates maintained were kept in triplicate, and zones around the wells were closely monitored.

## 3. Results

### 3.1. X-ray Diffraction Method (XRD) 

To confirm the structural properties of the synthesized cupric oxide nanoparticles (CuO NPs), bulk and exfoliated g-C_3_N_4_, and 30 wt%, 40 wt%, 50 wt%, 60 wt%, 70 wt% of CuO loaded g-C_3_N_4_ nanocomposites were evaluated by powder X-ray diffraction (P-XRD) analysis. The XRD pattern of g-C_3_N_4_, with hexagonal symmetry, is also presented in Figure 3A. The diffraction weak peak at 13.1° (110) and the strong peak at 27.4° (200) represent the g-C_3_N_4_ surfaces. Figure 3B represents the XRD spectra of the CuO nanoparticles. The diffraction spectra of pure CuO NPs at 32.35°, 35.26°, 39.35°, 48.97°, 53.15°, 58.90°, 61.78°, 66.35°, 68.88°, and 72.13° corresponded to the (-110), (111), (111), (202), (020), (202), (113), (004), (220), and (311) crystal facets and the lattice parameters a = 4.68 Å, b = 3.43 Å, c = 5.13 Å, β = 99.47°. Both diffracted peaks of the *g*-C_3_N_4_ and CuO NPs could be seen clearly for all different weight percentages of CuO-loaded *g*-C_3_N_4_ nanocomposites (Figure 3C).

### 3.2. Scanning Electron Microscopy (SEM)

The morphology and elemental composition of the products were analyzed through SEM and EDS. Figure 4 displays the SEM images of the obtained CuO sample from the dry synthesis method of the copper acetate precursor, the bulk *g*-C_3_N_4_, the exfoliated g-C_3_N_4_, and the CuO-loaded g-C_3_N_4_ nanocomposite. From Figure 4A,B, the sheet-like structure of bulk *g*-C_3_N_4_ (Figure 4A) and exfoliated *g*-C_3_N_4_ (Figure 4B) is observed. Moreover, thermal treatment resulted in a sheet-like *g*-C_3_N_4_ structure. Figure 4C–F shows the SEM images of the CuO nanoparticles and CuO loaded onto the *g*-C_3_N_4_ nanocomposites. This analysis confirms that the shape of as-prepared CuO NPs shows solid agglomeration, as seen in Figure 4C,D. Furthermore, the successful loading of CuO Nps on exfoliated *g*-C_3_N_4_ is confirmed in Figure 4E,F. The SEM-EDS measurements of the presence of C and N atoms in the nanosheets are shown in Figure 4G.

### 3.3. UV-Visible Diffuse Reflectance Spectroscopy (UV-Vis DRS)

The optical absorption properties of CuO, *g*-C_3_N_4_, and CuO-loaded *g*-C_3_N_4_ are presented in Figure 5. The estimated band gap is obtained by extrapolating the straight portion of (α*hv*)^2^ against the *hv* plot to the point α = 0, i.e., 1.10, 2.89, 1.17, 1.06, 1.03, 1.30, and 1.16 eV for the bare CuO, *g*-C_3_N_4_, 30 wt% CuO loaded *g*-C_3_N_4_, 40 wt% CuO loaded *g*-C_3_N_4_, 50 wt% CuO loaded *g*-C_3_N_4_, 60 wt% CuO loaded *g*-C_3_N_4_, and 70 wt% CuO loaded *g*-C_3_N_4_, respectively, as shown in Figure 5A–G. The UV-Vis spectrum of the sphere-like structured CuO nanoparticles (inset of Figure 5B) showed two absorptions at 290 and 355 nm. The *g*-C_3_N_4_ and 30 wt% CuO/*g*-C_3_N_4,_ 40 wt% CuO/*g*-C_3_N_4_, 50 wt% CuO/*g*-C_3_N_4_, 60 wt% CuO/*g*-C_3_N_4_, 70 wt% CuO/*g-*C_3_N_4_ manifest the absorption maxima in the region of 310 nm and 230–380 nm, respectively (inset of Figure 5A,C–G).

### 3.4. Antimicrobial Activity

The antibacterial activity of bare CuO, bulk *g*-C_3_N_4_, exfoliated *g*-C_3_N_4_, and CuO-loaded *g*-C_3_N_4_ was analyzed against various bacterial strains of Gram-positive bacteria (*Staphylococcus aureus* ATCC 25923, *Streptococcus pyogenes* ATCC 19615) and Gram-negative bacteria (*Escherichia coli* ATCC 25922, *Pseudomonas aeruginosa* ATCC 27853). The CuO NPs strains manifest a larger zone of inhibition, and a smaller zone of inhibition is manifested by the resistant strains. In regards to the zone of inhibition, the Gram-positive bacteria exhibited the highest activity toward CuO, while the Gram-negative bacteria manifested the lowest activity [1]. The inactivation rates of *Staphylococcus aureus* and *Streptococcus pyogenes* are not significantly different from each other; however, compared to that of *E. coli*, they are significantly higher. The antibacterial activity exhibited by CuO NPs and different weight percentages of 40 wt%, 50 wt%, 60 wt%, and 70 wt% cupric oxide-loaded g-C_3_N_4_ were effective against both Gram-positive and Gram-negative bacteria (Figure 6).

The effect of 70 wt% CuO-loaded *g*-C_3_N_4_ nanocomposites on the development of *Fusarium oxysporum* and *Trichoderma viride* as determined by the diameter of the fungal colonies on the samples, non-nanocomposite samples, and other samples of 1 mg/10 mL 70 wt% of CuO loaded *g*-C_3_N_4_ solution with the various concentrations of 20, 40, 60, 80 μL, respectively, after 4 days of incubation. In addition, Figure 7 shows the change in the diameter of fungal colonies for various concentrations of CuO/*g*-C_3_N_4_ solutions. These results show that 70 wt% of CuO-loaded g-C_3_N_4_ nanocomposite does not satisfactorily inhibit the fungal species.

## 4. Discussion

In this study, for the first time, cupric (II) oxide (CuO) nanoparticles loaded with exfoliated graphitic carbon nitride (*g*-C_3_N_4_) nanocomposite also showed antimicrobial activity. The current trending field of nanotechnology is critical and requires environmentally safe methods for the synthesis of nanoparticles. Here, a facile, rapid route and a low-cost approach for the preparation of stable cupric oxide (CuO NPs) nanoparticles and graphitic carbon nitride using a dry synthesis method is reported. The characteristics of the formed cupric (II) oxide loaded on graphitic carbon nitride nanocomposites were confirmed using XRD, UV-Vis (DRS), EDX, and SEM analyses.

The XRD patterns of CuO-loaded *g*-C_3_N_4_ nanosheets demonstrated their sheet-like structure [20,21]. The XRD patterns of CuO-loaded g-C_3_N_4_ nanocomposite revealed their crystalline structure. The XRD pattern of g-C_3_N_4_ was also presented in Figure 3A. The diffraction weak peak at 13.1° (110) and the strong peak at 27.4° (200) represent the g-C_3_N_4_ surface, arising from the in-plane structure of triazine, with the typical interplanar staking peaks of the inner layer structural packing. Figure 3B represents the XRD spectra of the CuO nanospheres particles. The diffraction spectra of pure CuO NPs at 32.35°, 35.26°, 39.35°, 48.97°, 53.15°, 58.90°, 61.78°, 66.35°, 68.88°, and 72.13° corresponded to the (-110), (111), (111), (202), (020), (202), (113), (004), (220), and (311) crystal facets of CuO [22]. In the diffractogram, all the strong and sharp peaks were in accord with the standard JCPDS card no 01-089-5895. The lattice parameters a = 4.68 Å, b = 3.43 Å, c = 5.13 Å, and β = 99.47° of the materials are indicated by a purely monoclinic phase and are additionally confirmed, without any phase impurity present in the synthesis materials. Both diffracted peaks of the *g*-C_3_N_4_ and CuO NPs could be seen clearly for all different weight percentages of the CuO-loaded *g*-C_3_N_4_ nanocomposites. The peak pertaining to g-C_3_N_4_ can be found as the mass ratio of *g*-C_3_N_4_ in CuO-loaded *g*-C_3_N_4_ nanocomposites of 30 wt%. While low intensities of g-C_3_N_4_ diffraction peaks are observed, this can be attributed to the low content of *g*-C_3_N_4_, which confirms the presence of *g*-C_3_N_4_ and CuO in the CuO-loaded *g*-C_3_N_4_ nanocomposites (Figure 3C).

The morphology and structure of bulk, exfoliated g-C_3_N_4_, CuO, and CuO-loaded *g*-C_3_N_4_ nanocomposites were analyzed by SEM. From Figure 3A,B, the sheet-like structure of bulk *g*-C_3_N_4_ (Figure 3A) and exfoliated *g*-C_3_N_4_ (Figure 3B) is clearly observed. Besides, thermal treatment resulted in a sheet-like *g*-C_3_N_4_ structure, and the layers became thinner, with a more detached and several-layer structure characteristic of two-dimensional exfoliated materials in the thermally exfoliated processed nanosheets. Furthermore, scanning electron microscopy analysis confirmed that the as-prepared CuO NPs exhibit nanoparticles of uniform size, as shown in Figure 3C,D. In CuO NPs loaded g-C_3_N_4_, Figure 3E,F shows that the nanoparticles were loaded onto the crumpled and rippled surface of the g-C_3_N_4_ sheets.

One of the most important electronic parameters for metal oxide nanomaterials is band gap energy. The band gap energy impacts the activity of the molecular adsorption sites and affects adsorption activity. The band gap energy for these samples was calculated from the optical absorption experiments using the Tauc equation, in which
(α*hv*) = *A* × (*hv* − *E*_g_)(1)
where the optical absorption coefficient is indicated by α. *A* is the constant, *hv* is an incident photon energy, and *Eg* is the energy gap, respectively. α can be replaced with the absorbance of the sample. The estimated band gap was obtained by extrapolating the straight portion of (α*hv*)^2^ against the *hv* plot to the point α = 0, i.e., 1.10, 2.89, 1.17, 1.06, 1.03, 1.30, and 1.16 eV for the samples of bare CuO NSs, *g*-C_3_N_4_, 30 wt% CuO loaded *g*-C_3_N_4_, 40 wt% CuO loaded *g*-C_3_N_4_, 50 wt% CuO loaded *g*-C_3_N_4_, 60 wt% CuO loaded *g*-C_3_N_4_, and 70 wt% CuO loaded *g*-C_3_N_4_, respectively, as shown in Figure 5A–G. The reason for the absorption was that the 3p state was located at the conduction band in the *g*-C_3_N_4_, resulting in a decrease in the band gap. Its band gap energy is 2.89 eV, as shown in Figure 5A. This indicates that *g*-C_3_N_4_ enhances the visible light utilization ability. The UV-Vis spectrum of the sphere-like structured CuO nanoparticles (inset of Figure 5B) showed two absorptions at 290 and 355 nm. The *g*-C_3_N_4_ and 30 wt% CuO/*g*-C_3_N_4_, 40 wt% CuO/*g*-C_3_N_4_, 50 wt% CuO/*g*-C_3_N_4_, 60 wt% CuO/*g*-C_3_N_4_, and 70 wt% CuO/*g*-C_3_N_4_ manifest the absorption maxima in the region of 310 nm and 230–380 nm, respectively (inset of Figure 5A,C–G).

The antibacterial activity of various nanoparticles for various bacteria using a well diffusion assay is shown in Figure 6A–D. The results illustrated an increase in CuO weight percentage directly proportionally to the antibacterial activity. The CuO NPs strains manifest a larger zone of inhibition, and a smaller zone of inhibition is manifested by the resistant strains. In regards to the zone of inhibition, the Gram-positive bacteria exhibited the highest activity toward CuO, while the Gram-negative bacteria manifest the lowest activity. The inactivation rates of *Staphylococcus aureus* and *Streptococcus pyogenes* are not significantly different from each other; however, compared to that of *E. coli*, they are significantly higher. These results indicated that Gram-positive bacteria are more resistant to the antibacterial activity of CuO-loaded *g*-C_3_N_4_ than Gram-negative bacteria, such as *Escherichia coli* and *Pseudomonas aeruginosa* [21]. The antibacterial activity results for CuO-loaded *g*-C_3_N_4_ indicated a superior resistance to Gram-positive bacteria compared to Gram-negative bacteria such as *Escherichia coli* and *Pseudomonas aeruginosa* [1,23,24,25]. These results demonstrated that susceptibility to inactivation is dependent upon the microorganism, as shown in Table 1. Future work is necessary to draw firm conclusions about the improved *g*-C_3_N_4_ inactivation performance for Gram-negative over Gram-positive microorganisms.

The photocatalytic activities are depended on the amount of active radical species (such as h^+^, e^-^, **·**OH, **·**O_2_^-^) produced in the reactions [21]. Previous studies confirmed that radicals play important roles in photoreactions. Figure 8 shows an increase in active metal oxide, indicating that *g*-C_3_N_4_ under light irradiation can easily produce **·**OH, due to the oxidation between H^+^ and H_2_O. Subsequently, the **·**OH reacts with the bacterial cell membrane and cleaves the linkages. It is commonly known that *g*-C_3_N_4_ is a photocatalytic agent; therefore, the application of light and well-produced radicals, such as **·**O_2_^-^ and **·**OH, play important roles in the photocatalytic process [14,15,16,20,26]. This compound is highly active under light irradiation, but in this study, dark conditions were maintained. However, it is still moderately active against Gram-positive and Gram-negative bacteria, compared to metal oxide nanoparticles [27,28], but CuO/*g-*C_3_N_4_ nanocomposites exhibit good antibacterial activity. Finally, the bacteria cell was distorted. If its cell membrane is damaged, the intracellular contents (such as DNA, protein, and mitochondria) can leak from the extracellular suspensions of the cell, resulting in the destruction of the cell structure.

In previous reports, *g*-C_3_N_4_ exhibited excellent activity in visible light irradiation due to the production of **·**O_2_^-^ and **·**OH active radical species in photocatalytic disinfection [27]. The *g*-C_3_N_4_ exhibits good photoelectric properties and band structure. In our case, however, since this study was performed under room conditions in vitro, the results showed that fungal growth was not inhibited [21,29,30]. Thus, future studies should be conducted in such a way that a fungal species should be subjected to photocatalytic irradiation [1]. Here, this study was not performed in the presence of sunlight. Under lab conditions, the pathogenic fungi (*Fusarium oxysporum* and *Trichoderma viride*) were allowed to interact with 1 mg/mL in different concentrations of 20, 40, 60, and 80 μL (triplicates), and the compounds at these particular concentrations did not inhibit the fungi in the presence of this CuO/*g*-C_3_N_4_ nanocomposite [31,32]. This is because *g*-C_3_N_4_ masks the effect of the CuO molecules and does not inhibit fungi. According to Lin et al., *g*-C_3_N_4_ responds to nitrogen fixation and acts as a foliar fertilizer [26]. Since effective nitrogen fixation results in prominent nitrogen fixations for a good yield, as well as predominant protein synthesis, the result is the growth of fungi, rather than fungal inhibition. Therefore, further studies should be conducted in such a way that the CuO/*g*-C_3_N_4_ compounds are subjected to photocatalytic exposure in the presence of sunlight. In this way, the inhibition of fungi may be possible for these molecules.

## 5. Conclusions

The various biological processes, including apoptosis, anti-angiogenesis, oxidative stress, chemotherapy, and inflammation, are modulated by inorganic nanoparticles. In this study, cupric (II) oxide (CuO) nanoparticles loaded with exfoliated graphitic carbon nitride (*g*-C_3_N_4_) nanocomposite assisted in the suppression or inhibition of bacterial growth and the interaction of fungi. The current trending field of nanotechnology is critical and requires environmentally safe methods for the synthesis of nanoparticles. Here, a facile, rapid route and a low-cost approach for the preparation of stable cupric oxide (CuO) nanospheres and graphitic carbon nitride using a dry synthesis method was reported. The characteristics of the formed copper (II) oxide loaded on graphitic carbon nitride nanocomposites were confirmed using XRD, UV-Vis (DRS), EDX, and SEM analyses. The XRD patterns of CuO-loaded *g-*C_3_N_4_ nanosheets demonstrated the sheet-like structure. The XRD patterns of CuO-loaded *g*-C_3_N_4_ nanocomposite revealed the crystalline structure. The SEM images of the synthesis CuO-loaded *g*-C_3_N_4_ nanocomposite morphology shows a two-dimensional sheet-like structure. The antibacterial activity was exhibited by CuO NPs, with different weight percentages of 40 wt%, 50 wt%, 60 wt%, and 70 wt% cupric oxide-loaded *g*-C_3_N_4_ against both Gram-positive and Gram-negative bacteria. The synthesized bulk *g*-C_3_N_4_, exfoliated *g*-C_3_N_4_, and lower weight percentages of the nanocomposite (30 wt%) showed low antibacterial activity. The optimum dose of 70wt % CuO-loaded *g*-C_3_N_4_ showed high antibacterial activity but did not exhibit satisfactory activity in antifungal studies conducted in dark fields. This is because *g*-C_3_N_4_ masks the effect of CuO molecules and does not inhibit fungi. The scope of using CuO-loaded *g*-C_3_N_4_ nanocomposites as antimicrobials needs to be further explored in the presence of sunlight.

## Figures and Tables

**Figure 1 jof-09-00310-f001:**
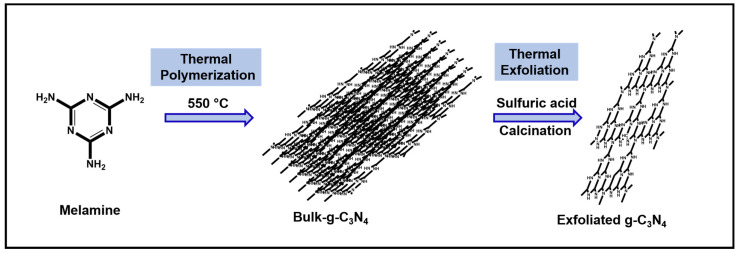
Simplified representation of the production of exfoliated *g*-C_3_N_4._

**Figure 2 jof-09-00310-f002:**
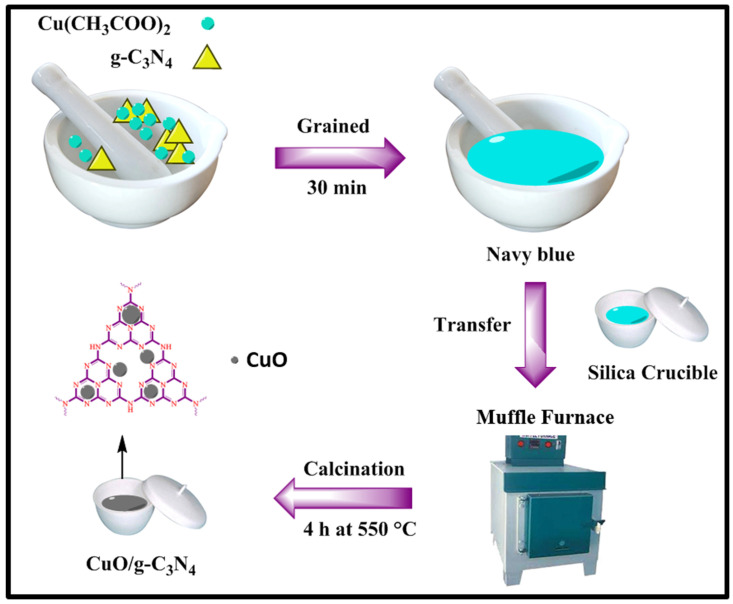
CuO/g-C_3_N_4_ nanocomposites prepared by dry synthesis method.

**Figure 3 jof-09-00310-f003:**
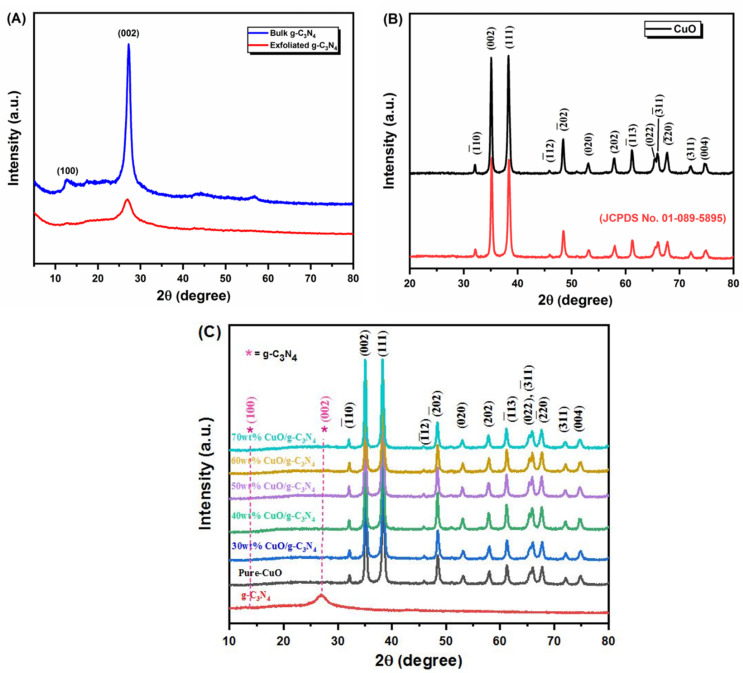
XRD diffractogram of (**A**) bulk *g*-C_3_N_4_ and exfoliated *g*-C_3_N_4_, (**B**) bare CuO and standard peak, (**C**) bare CuO and exfoliated *g*-C_3_N_4_, and various weight percentages of CuO loaded *g*-C_3_N_4_.

**Figure 4 jof-09-00310-f004:**
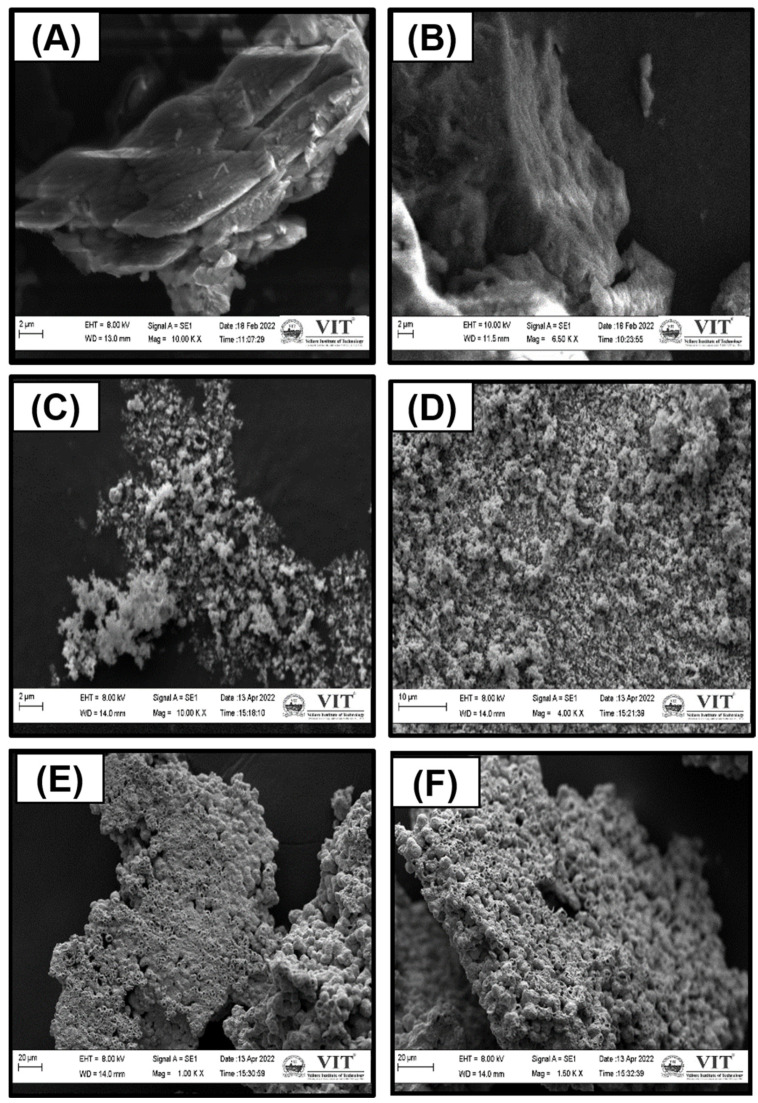
SEM figure of (**A**) bulk *g*-C_3_N_4_, (**B**) exfoliated *g*-C_3_N_4_, (**C**,**D**) CuO, (**E**,**F**) CuO/*g*-C_3_N_4_ nanocomposites, and (**G**) EDS spectra of bulk *g*-C_3_N_4_.

**Figure 5 jof-09-00310-f005:**
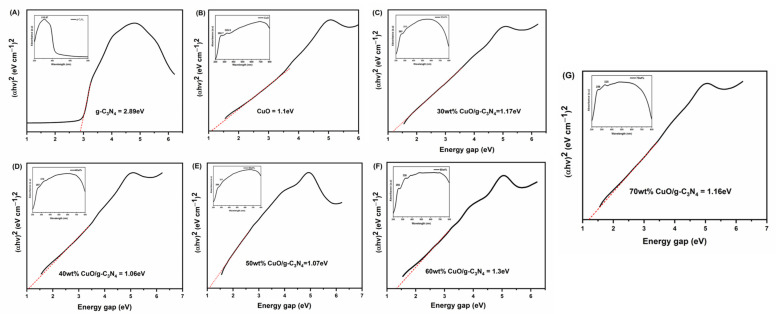
UV-DRS spectra exhibit band gap energy of exfoliated *g*-C_3_N_4_ (**A**), CuO (**B**), and CuO loaded *g*-C_3_N_4_ (**C**–**G**); (inset: absorption peak). (Bold black line represented by UV-DRS and dotted line noted by bandgap measurement line).

**Figure 6 jof-09-00310-f006:**
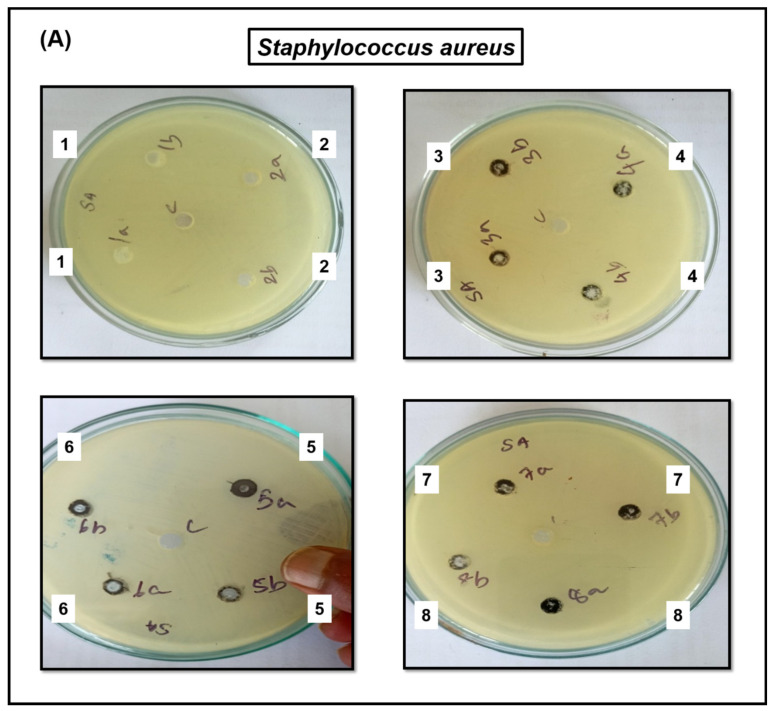
Antimicrobial activity of bulk *g*-C_3_N_4_ (1), exfoliated *g*-C_3_N_4_ (2), CuO (3), and different weight percentages of CuO loaded *g*-C_3_N_4_ nanocomposite, 30 wt% (4), 40 wt% (5), 50 wt% (6), 60 wt% (7), and 70 wt% (8) against *Staphylococcus aureus* (**A**), *Streptococcus pyogenes* (**B**), *Escherichia coli* (**C**), and *Pseudomonas aeruginosa* (**D**).

**Figure 7 jof-09-00310-f007:**
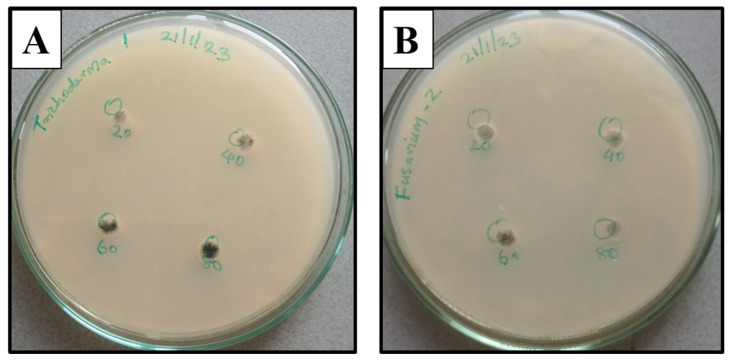
Antifungal activity of 70 wt% of CuO loaded g-C_3_N_4_ nanocomposite solution with the various concentrations of 20, 40, 60, and 80 μL (**A**), *Trichoderma viride*, and (**B**) *Fusarium oxysporum*.

**Figure 8 jof-09-00310-f008:**
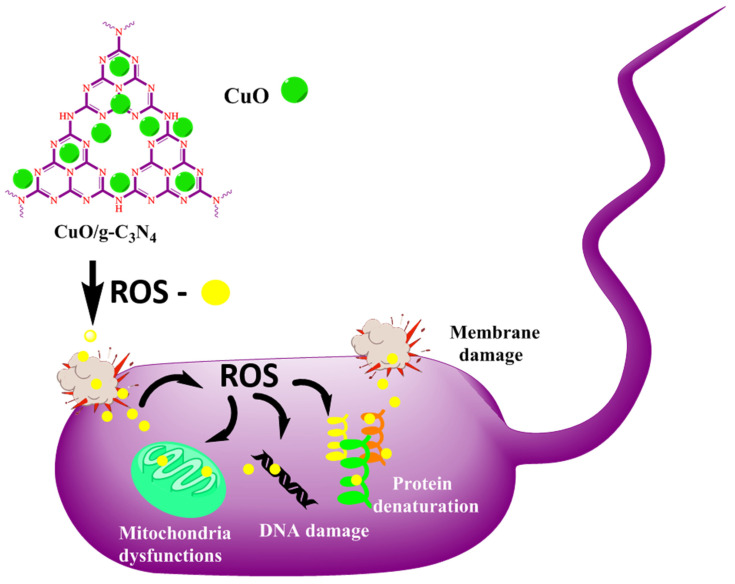
Plausible mechanisms of antimicrobial activity.

**Table 1 jof-09-00310-t001:** Antimicrobial activity of cupric oxide/g-C_3_N_4_nanocomposite.

Samples	*S. aureus* (mm)	*S. pyogenes* (mm)	*E. coli* (mm)	*P. aeruginosa* (mm)
Bulk-*g*-C_3_N_4_	R	R	R	R
Exfoliated *g*-C_3_N_4_	R	R	R	R
CuO	8	12	6	8
30 wt% CuO/*g*-C_3_N_4_	R	R	R	R
40 wt% CuO/*g*-C_3_N_4_	16	16	10	12
50 wt% CuO/*g*-C_3_N_4_	16	16	10	14
60 wt% CuO/*g*-C_3_N_4_	18	18	12	14
70 wt% CuO/*g*-C_3_N_4_	18	18	12	14

(R)—resistant.

## Data Availability

Not applicable.

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
