# Peer review of "Facile Synthesis and Characterization of Cupric Oxide Loaded 2D Structure Graphitic Carbon Nitride (g-C3N4) Nanocomposite: In Vitro Anti-Bacterial and Fungal Interaction Studies"

_jof, 2023, doi:10.3390/jof9030310_

Round 1

Reviewer 1 Report

After careful reading, I think this manuscript is well structured and written. However, there are few considerations to published in peer reviewed journal.

Introduction:

The introduction section is well organized

Methodology:

The introduction section is well written

Results:

The result section should be streamlined. The reader has a very difficult time taking in all of the results. I’m further suggesting reducing the content which already depicted on the figures and tables in your text and presenting only the main trends that are serving in this purpose.

Discussion:

This is interesting and can be something you could contribute to this field. The discussion especially needs to be reviewed and edited. Some information added in this section is baseless. Please re-consider within authors what you really wanted to clarify on this study and what was currently lacking in the scientific field based on careful literature review but sorry, I cannot see those from the current form.

General issues:

Detailed suggestions in attached zip file

A more minor problem was that the paper was difficult to follow, I believe mostly due to problems with heavy contents of result and logical flow of discussion section. Both sections especially need to be reviewed and edited.

Overall, I suggest the authors work on streamlining the results and discussion sections, and getting assistance in writing the results and discussion more clearly.  I think the data themselves are original, valuable, and should be published after some of these items are addressed by the authors.

Reference list:

Check again I can see several mistakes 

Reviewer 2 Report

It would have been better to present a table that includes the effect of different concentrations on the growth of the tested fungi, such as those that were exposed to bacteria
